# Quality evaluation of ground improvement by deep cement mixing piles via ground-penetrating radar

Hongyan Shen [1,2] ✉, Xinsheng Li [3], Ruifeng Duan [3,4] ✉, Yong Zhao[5], Jing Zhao[1,2], Han Che[1,2], Guoxin Liu[1,2], Zhijia Xue[6], Changgen Yan[6], Jiwei Liu[6], Chao Jiang[7], Boke Li[3], Hong Chang[3], Jianqiang Gao[6] & Yueying Yan[1] ✉

Deep cement mixing piles are a key technology for treating settlement distress of soft soil subgrade. However, it is very challenging to accurately evaluate the quality of pile construction due to the limitations of pile material, large number of piles and small pile spacing. Here, we propose the idea of transforming defect detection of piles into quality evaluation of ground improvement. Geological models of pile group reinforced subgrade are constructed and their ground-penetrating radar response characteristics are revealed. We have also developed ground-penetrating radar attribute analysis technology and established ground-penetrating radar technical system for evaluating the quality of ground improvement. We further prove that the ground-penetrating radar results integrating single-channel waveform, multi-channel section and attributes can effectively detect the defects and stratum structure after ground improvement. Our research results provide a rapid, efficient and economic technical solution for the quality evaluation of ground improvement in soft soil subgrade reinforcement engineering.

Subgrade settlement is a serious distress problem faced by the construction of highways and railways[1–7]. It is particularly prominent in soft loess covered areas, river valley sections, mined out areas, karst development areas, etc., which poses a huge threat to the service life and operation safety of roads. Therefore, in the process of subgrade construction, it is necessary to take effective measures by every means to improve the quality of subgrade construction to prevent the occurrence of subgrade settlement distress[8–11]. Deep cement mixing (DCM) pile is a kind of cement plus solid, which uses cement as the main curing agent and forcibly mixes the soft soil and cement paste in the deep part of the foundation through mixing machinery, making the soft soil harden into a cement reinforced soil with integrity and certain strength, thereby improving the strength of the foundations[12–16]. It has the characteristics of low specific gravity of consolidated body, large bearing capacity, low permeability coefficient, and has the advantages of low cost, short construction time and maximum utilization of the original soil. Therefore, it has been widely used in soft soil ground improvement, foundation settlement treatment and other engineering construction around the world[17–24]. At present, DCM piles have become one of the key technologies in highway distress prevention and treatment engineering. However, some piles may have different degrees of defects due to the dynamic conditions of soil layer and groundwater, construction technical defects, construction loopholes and other reasons. It has laid hidden dangers for inducing accidents[25,26]. Therefore, pile defect detection and pile

[1]School of Earth Sciences and Engineering, Xi'an Shiyou University, Xi'an 710065, P.R. China. [2]Shaanxi Key Laboratory of Petroleum Accumulation Geology, Xi'an 710065, P.R. China. [3]Shaanxi Geo-mining Geophysical and Geochemical Exploration Team Co. Ltd., Xi'an 710043, P.R. China. [4]National Engineering Research Center of Offshore Oil and Gas Exploration, Beijing 100028, P.R. China. [5]Shaanxi Land Construction Surveying, Planning and Design Institute Co. Ltd., Shaanxi Land Construction Group, Xi'an 710075, P.R. China. [6]School of Highway, Chang'an University, Xi'an 710064, P.R. China. [7]Gansu Luqiao Highway Investment Co., Ltd., Lanzhou 730030, P.R. China. ✉e-mail: shenhongyan@xsyu.edu.cn; 450991137@qq.com; Yanyueying@xsyu.edu.cn

construction quality evaluation are important guarantee technologies for subgrade distress treatment.

The traditional engineering pile foundation detection mainly adopts static load testing, drilling coring, high-strain and low-strain dynamic measurement. Static load testing[27–29] and drilling coring[30,31] are relatively accurate and fair methods for pile foundation detection. However, both of these methods are destructive testing techniques with long cycles, high costs, low sampling ratios (1-2%) and poor representativeness of sampling. High-strain dynamic measurement[32–34] is to impact the pile top with a heavy hammer, and then analyze and determine the vertical compressive bearing capacity and pile integrity of a single pile based on the wave theory. This method can only provide reference bearing capacity. Low-strain dynamic measurement[35,36] involves applying a dynamic force (dynamic load) to the pile top, and the structural integrity of the pile body is judged through observation and analysis of dynamic response signals. This method has the advantages of fast detection speed, low cost and good reliability, and has a good detection effect for high-strength pile foundations, such as cast-in-place piles, prefabricated piles, etc. Currently, pile foundation testing is mainly based on this technology[37–39]. However, DCM piles are locally sourced and mixed with cement paste to form piles. Although there are differences between the elasticity of the piles and the original stratum, the difference is far less obvious than that of the piles cast with sand, cement and coarse aggregates as the main material[13,29]. Therefore, the low-strain dynamic measurement technology developed on the basis of elastic theory may not be suitable for the detection of DCM piles. In addition, the number of DCM pile groups is too huge, the pile spacing is small, and the pile group density is high for the soft soil reinforcement engineering. It is difficult to obtain obvious detection effect and high economic benefits by using this traditional pile foundation testing technology.

Ground-penetrating radar (GPR) is a survey technology based on electromagnetic field theory. Its basic principle is to transmit ultra-high frequency ($10^6$-$10^9$ Hz) pulse electromagnetic waves, and then analyze and infer the structure and physical properties of underground media according to the changes of echo amplitude, waveform, frequency and other characteristics by using the differences of electromagnetic properties of underground media[40–43]. It has the advantages of high resolution, intuitive results and fast scanning speed. In recent years, it has become an important technology and means for various engineering quality detection, near- surface stratum structure and defect survey[44–55]. Recent research has shown that there are significant differences in electromagnetic properties between DCM pile and original stratum, and there are also significant differences in electromagnetic properties between defective and complete piles[56,57]. Therefore, GPR technology is fully applicable to the quality inspection of DCM piles.

Based on the above reasons, we proposed to use GPR technology to detect the defects of DCM piles, and then realize the quality evaluation of ground improvement. Our innovations are to put forward the idea of transforming defect detection of pile foundations into quality evaluation of ground improvement, construct geological models of subgrade reinforced by pile groups and reveal theirs GPR response characteristics, develop a GPR multi-attribute information extraction technology, and establish a GPR technical system for evaluating the quality of ground improvement. The pile length, strong/weak reinforcement stratum, primary stratum, the buried depth and range of defective formation can be effectively identified by integrating the GPR information of single-channel waveform, multi-channel section and attributes. Our research results provide a fast, efficient and economic way for the quality evaluation of DCM pile reinforcement for soft soil foundation, and fill the technical gap of efficient detection of pile foundation defects in the case of large number of pile groups with small pile spacing and high pile group density. Moreover, our technology can also be expanded for pile foundation defect detection and

pile construction quality evaluation in other projects under high-density pile groups, such as slope protection, dam seepage prevention, foundation reinforcement, and other engineering.

## Results

### Geological models and their GPR response characteristics

The purpose of using DCM piles in soft soil foundation is to enheance the bearing capacity of the stratum and then to improve the stability of the foundation. If the subgrade strengthened by DCM pile groups is regarded as a complete set of strata, the problem of pile foundation detection can be transformed into the problem of ground improvement quality evaluation (Fig. 1a, b). If there are quality problems in the foundation evaluation, it will also reflect that there are defects in the pile foundations. The purpose of this is to create conditions for the selection of continuous detection technologies. It can significantly reduce the cost of pile foundation detection while effectively improving the efficiency and effect of pile foundation testing.

In order to effectively use GPR to detect the quality of pile foundations or evaluate the effectiveness of ground improvement, it is necessary to first understand the propagation laws of electromagnetic waves under different geological conditions of subgrade and the GPR response characteristics. For this reason, we analyzed the electromagnetic waves propagation and GPR response characteristics of the geological models of reinforced subgrade by pile groups. In the foundations without pile (Fig. 1c), electromagnetic waves are reflected only at the wave impedance interfaces (stratum interfaces). For the foundations reinforced by complete piles (Fig. 1d), in addition to forming reflections at the interfaces of wave impedances, electromagnetic waves also generate complex reflection, transmission, refraction and guided waves between piles as well as between piles and the original stratum. Thus, GPR response characteristics with relatively stable frequency and amplitude are formed in the equivalent formation. For the foundations reinforced by defective piles (Fig. 1e), due to defects in the piles or stratum (such as pile breaking, necking, segregation, etc.), the energy distribution of electromagnetic waves at the defect locations will be changed, which will also lead to change in GPR response characteristics. Specifically, the amplitude become weaker (type I response characteristics in Fig. 1e) or stronger (type II response characteristics in Fig. 1e), and the frequency also become lower or higher.

In the quality evaluation process of ground improvement, although the subgrade strengthened by piles is equivalent to a set of strata, the GPR response characteristics of equivalent strata are completely different from those of strata without pile, and there may be significant differences between the GPR response characteristics of defective piles and complete piles (Fig. 1c, d, e). Therefore, these GPR response characteristics and differences become an important basis for identifying whether piles have defects and evaluating the quality of ground improvement.

### Technical system

The outstanding characteristics of ground improvement by DCM piles are that the project progress is fast, the construction process is closely connected, and the time available for pile foundation defect detection and ground improvement quality evaluation is limited. It is required that pile foundation defect detection and ground improvement quality evaluation not only to quickly and accurately detect the distribution areas and scopes of defective piles or unfavorable stratum, but also to meet the needs of rapid construction of highway engineering, so as to timely feedback on the detection results and take timely measures to supplement piles. Therefore, we have established a technical system for using GPR to detect defects in DCM piles and evaluate the quality of ground improvement based on engineering geological survey data and highway construction site conditions (Fig. 2). Specifically, it includes the following four aspects.

**Geological basis**. Comprehensively master the stratum structure and physical properties of subgrade as well as the thickness of soft soil layer based on engineering geological survey data. Deeply understand design the parameters such as pile material, diameter, length and spacing as well as the technical scheme for subgrade construction.

**Data acquisition**. Design GPR survey lines or networks based on the site conditions of highway construction. The basic principle is to lay survey lines along the extension direction of the highway and along the axis of the pile foundations (Fig. 1b). For pile foundation detection with a length of several meters to more than ten meters, it is recommended to select 100-200 MHz antennas and use profiling method to continuously collect GPR data. Note that the sampling length must completely include the equivalent layer depth of the reinforced subgrade by pile foundations, and the sampling rate is generally from 0.1 to 0.4 ns to meet the detection requirements.

**Data processing**. Carry out fine processing of GPR data around key links such as energy enhancement in deep layers, interference noise suppression and multiple attenuation. Ensure that GPR data processing results with high signal-to-noise ratio, high fidelity and high resolution are obtained.

**Data interpretation**. Integrate single-channel waveform, multi-channel section and attribute information, and combine drilling results to interpret geological information in GPR data. Realize the detection of pile foundation defects and accurate quality evaluation of ground improvement. It should be emphasized here that 1-2 boreholes can be properly arranged on the GPR survey line (or network), rather than many boreholes. The purpose of borehole coring is not only to verify the GPR interpretation results, but also to calibrate the GPR section with borehole results to further calculate and obtain more accurate electromagnetic wave velocities to achieve time-depth conversion processing of the GPR section.

## Project overview and GPR data acquisition

At present, DCM piles have been widely used in the treatment of subgrade settlement distress caused by soft loess settlement[24,58–60]. We have carried out the application and practice of GPR technology in defect detection of DCM piles and quality evaluation of ground improvement based on the construction project of Anlin highway in Gansu Province, China. Anlin highway stats from Anjiazui Village, Taishi Town, Linzhao County, and ends at Baichuan Village, Dongyuan Township, Linxia County (Fig. 3a), with a total length of 56.7 km. It would be constructed according to the technical standards of two-way four lane second-class highway.

It was found that there were signs of soft soil foundations in Sanjiaji section during the highway survey stage. A combination of drilling and excavation exploration was used to obtain foundation soil samples. It was found by observing the foundation soil samples that the undisturbed soil particles in this section were mainly composed of silt particles (particle size 0.006-0.078 mm), with obvious porosity. Macropores and wormholes were visible to the naked eye, and contained a large mount of salt crystals. Therefore, it was determined that this section belongs to typical soft loess. DCM pile technology was selected to strengthen and treated the subgrade of this section in order to ensure high-quality construction of highway engineering. The design scheme for subgrade reinforcement is shown in Supplementary Fig. S1. The DCM piles were arranged in a regular quincunx pattern, with a pile diameter of 0.5 m, a pile spacing of 1.5 m, and a pile length of 10.0 m. A total of more than 10,000 DCM piles were built in the entire highway section.

28 days after the completion of pile construction, the work of using GPR technology to evaluate the quality of pile foundations and

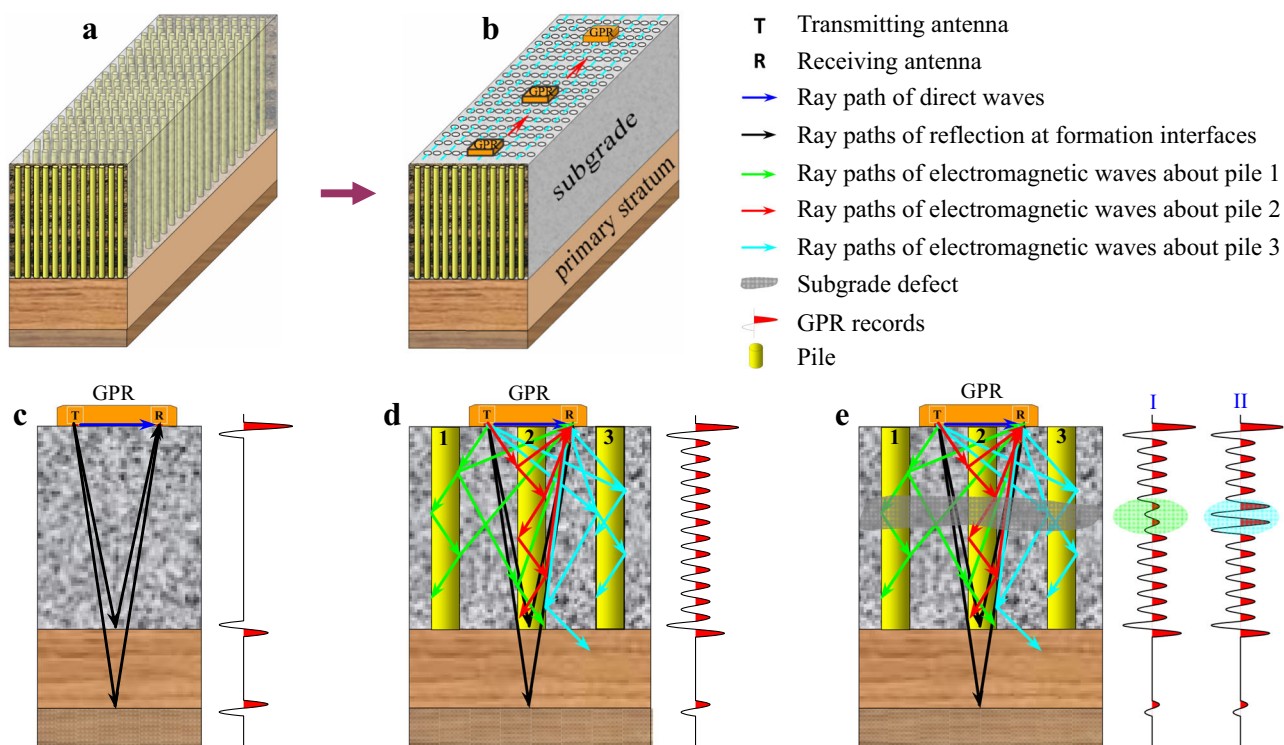

**Fig. 1 | Geological models of ground improvement by pile groups and their GPR response characteristics. a** Geological model of subgrade reinforced by pile groups. **b** Equivalent geological model of subgrade reinforced by pile groups. Pile groups are equivalent to a stratum, and defect detection of pile foundations is transformed into quality evaluation of ground improvement. **c** Stratum without pile, (**d**) stratum with complete piles, (**e**) stratum with defective piles and theirs GPR response characteristics. Type I denotes energy weakening, and type II represents energy enhancement.

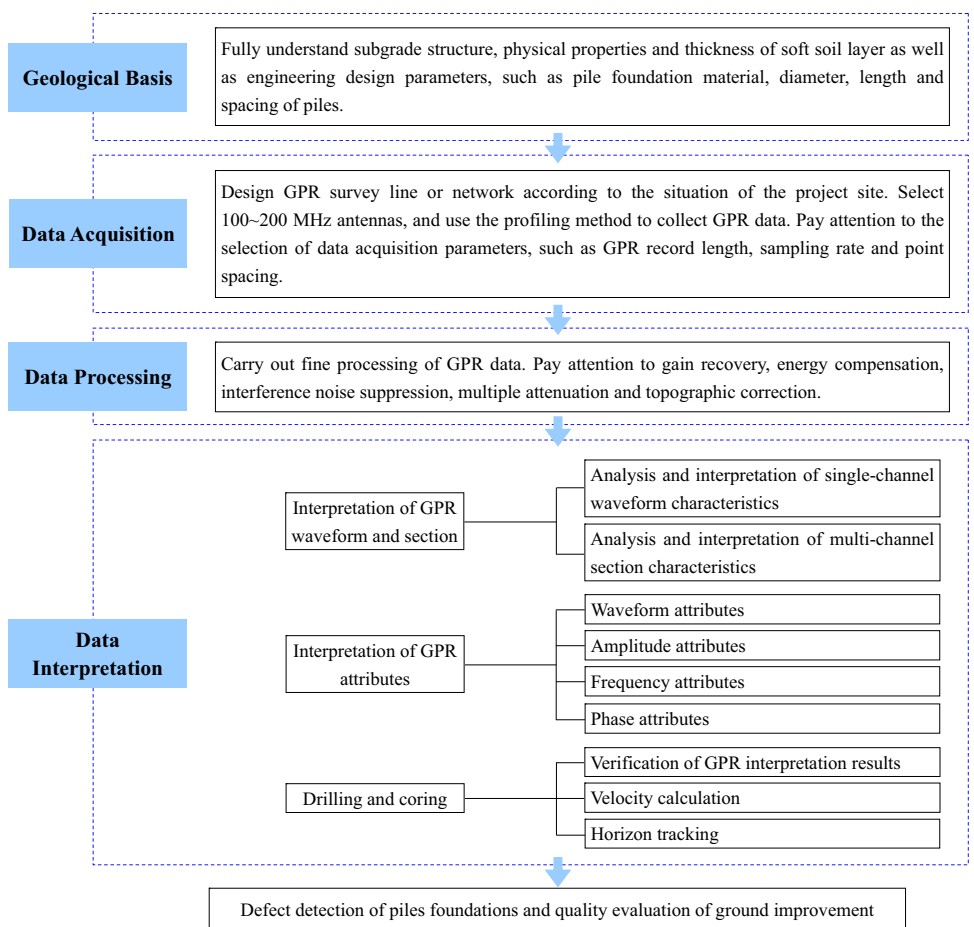

Fig. 2 | Technical system for DCM pile foundation defect detection and ground improvement quality evaluation via GPR. The technical system includes four parts that are geological basis, data acquisition, data processing, and data interpretation.

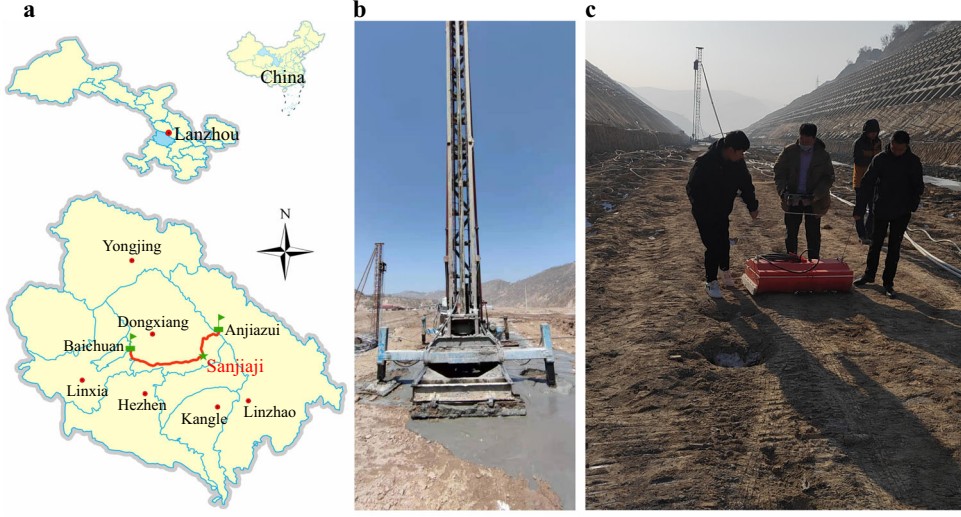

Fig. 3 | Location of work area and field data acquisition. a Location of work area. b DCM pile construction scene. c GPR data acquisition scenario.

the effect of subgrade reinforcement was carried out. We used the SIR-4000® GPR system, selected 100 MHz antennas, and used the profiling method for GPR data acquisition. The scene of piling and field GPR data acquisition is shown in Fig. 3b, c. The data acquisition parameters included that the sampling rate was 0.351 ns, the sampling length was 1024 points per channel, and the channel spacing was approximately 2.0 cm.

## Interpretation of GPR section

Taking a survey line in the exploration area as an example, the application effect of our proposed technology is introduced in detail. Supplementary Fig. S2 shows the original GPR data obtained for this survey line and Fig. 4a shows the GPR data processing results according to the data processing flow in Supplementary Fig. S3. Four typical channels were extracted from the GPR section in Fig. 4a for

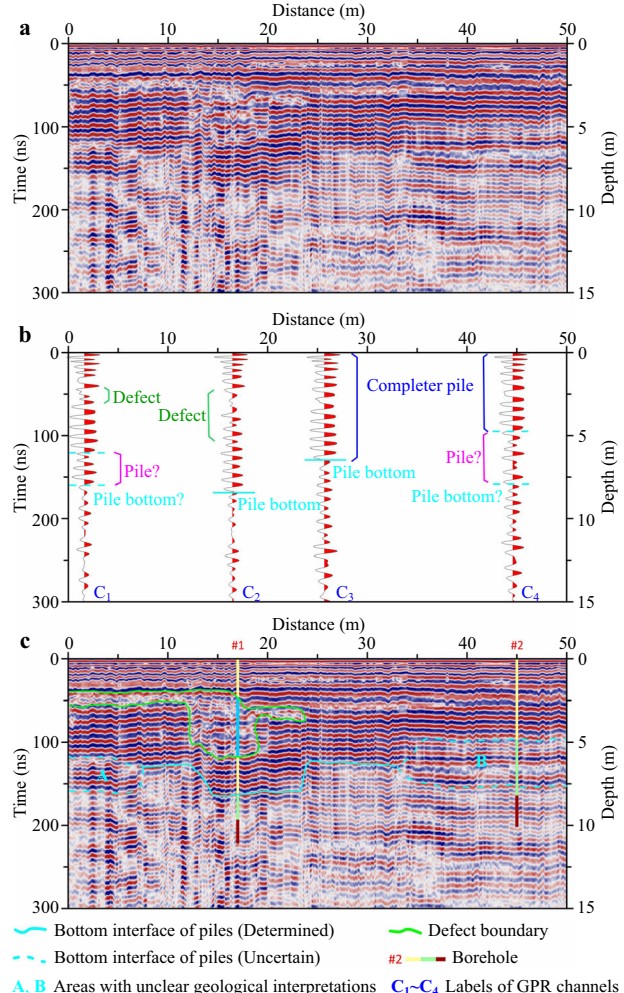

**Fig. 4 | Interpretation of GPR section. a** GPR data processing results. **b** Waveform interpretation of typical channels. **c** Multi-channel section interpretation. Interpretation of GPR section is performed after analyzing and interpreting the waveform characteristics of typical channels. It is not possible to determine whether pile foundations exist in A and B areas. Source data are provided as a Source Data file.

waveform feature analysis and interpretation in order to accurately grasp the GPR response characteristics. The extracted GPR channels were identified as $C_1$, $C_2$, $C_3$ and $C_4$, respectively (Fig. 4b). In the GPR channel marked with $C_1$, the energy between 44 and 57 ns was weak and the waveform changed irregularly. It was interpreted that there were subgrade (or pile foundation) defects between 44 and 57 ns of the channel based on the analysis conclusion in Fig. 1e. The energy between 122 ns and 163 ns of the channel was also weak, but a wavelet with strong energy appeared near 163 ns. Therefore, it was impossible to judge whether the pile bottom was at 122 ns or 163 ns based solely on the waveform characteristics. In the GPR channel labeled $C_2$, there was waveform disorder between 48 and 119 ns. This indicated that there were defects in the subgrade (or pile foundation). The GPR energy suddenly weakened at 163 ns of this channel, which should be the GPR response characteristic of the pile bottom. In the GPR channel labeled $C_3$, the waveform changed were more regularly. It was judged that the pile foundation was complete or there was no defect in the subgrade based on the analysis conclusion in Fig. 1d. The amplitude attenuation of the channel appeared at 131 ns, indicating that this might be the pile bottom. In the GPR channel marked with $C_4$, there was a weak and chaotic energy phenomenon between 97 and 159 ns, so it was impossible to determine whether the pile bottom was at 97 ns or 159 ns.

A preliminary interpretation of the GPR section can be further made based on the above single-channel waveform interpretation results, and combined with the variation characteristics of waveform, amplitude, frequency and phase of the section. The interpretation results are shown in Fig. 4c, in which the area surrounded by a green irregular circle was interpreted as a subgrade defect, the cyan solid line was interpreted as a clear bottom interface of subgrade reinforcement, and the cyan dotted lines were interpreted uncertain bottom interfaces of subgrade reinforcement.

### Interpretation of GPR attributes
Although the quality of DCM piles and the effect of subgrade reinforcement can be preliminarily evaluated based on the single-channel waveform and multi-channel section characteristics of GPR data, some interpretation conclusions are ambiguous because the information carried by GPR data has not been fully explored. As shown in Fig. 4c, it is difficult to give an accurate conclusion as to whether there are pile foundations or defects in the left A and right B areas of the section. Therefore, we used the GPR attribute information extraction and analysis technology that we recently developed to further process the section in Fig. 4a. It was attempted to clearly reveal whether the pile foundation was complete and accurately evaluated the quality of weak subgrade reinforcement through GPR attribute information. Here, we only selected one of the most representative GPR attributes from waveform, frequency, amplitude and phase attributes categories for in-depth analysis and geological information interpretation.

**Arc length of time window (AL).** There must be differences in GPR response characteristics between reinforced and primary stratum, between strong and weak reinforced stratum as well as between defective and non defective stratum. However, when the differences in GPR response characteristics are not obvious, it is difficult to distinguish them in conventional GPR sections, such as the left A and right B areas in Fig. 4c. It could be seen from the AL attribute in Fig. 5a that the boundary between strong and weak values clearly showed the bottom interface of the strongly reinforced stratum (cyan solid line). The interface was interpreted as a pile bottom interface in the GPR section in Fig. 4c, which might be a wrong interpretation results. In addition, the energy disturbance areas of this attribute section clearly delineated the defective areas of the subgrade (irregular green solid line circles), which was clearer and more accurate than that in the GPR section in Fig. 4a.

**Product of instantaneous amplitude and cosine of instantaneous phase (PIACIP).** It could see from the PIACIP attribute in Fig. 5b that the effect was equivalent to that of the AL attribute in Fig. 5a. The bottom interface of the strongly reinforced stratum was clear (cyan solid line) as well as the defect areas of the subgrade were also clear (irregular green solid line circles). It further confirmed that the interpretation of the AL attribute was reliable.

**Slope of instantaneous frequency (SIF).** As could be seen from Fig. 5c that the bottom interface of the strongly reinforced layer was not reflected in the section of SIF attribute, but it clearly revealed the interface between the weakly reinforced layer and the primary stratum (bottom interface of the weakly strengthened layer, purple solid line). The bottom interface information of the weakly reinforced layer could not be obtained from the above two attributes in Fig. 5a, b and the conventional GPR section in Fig. 4a. The disordered areas of this attribute also clearly circled the defective areas of the subgrade (irregular green solid line circles).

**Slope of reflection strength (SRS).** We could see from Fig. 5d that the SRS attribute also accurately identified the boundary between the weakly reinforced layer and the primary stratum (purple solid line),

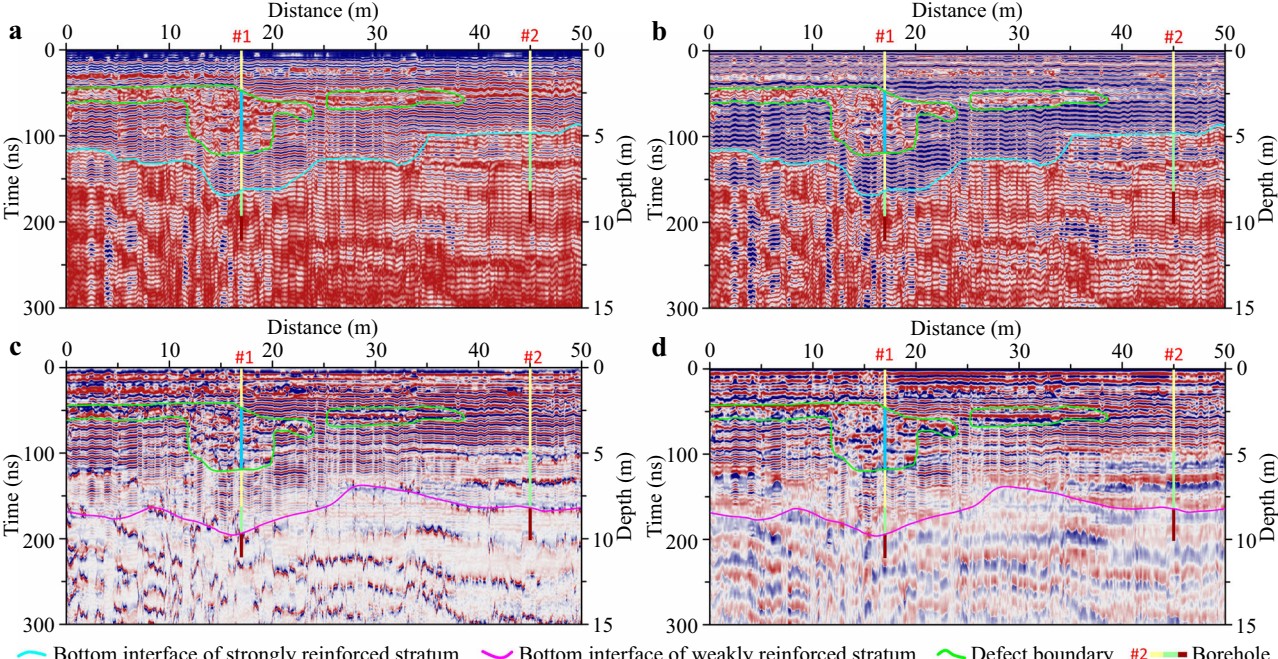

**Fig. 5 | Interpretation of GPR attributes. a** Arc length (AL) attribute with an analysis time window of 1.0 ns. **b** Product of instantaneous amplitude and cosine of instantaneous phase (PIACIP) attribute. **c** Slope of instantaneous frequency (SIF) attribute. **d** Slope of reflection strength (SRS) attribute. The bottom interface of the strongly reinforced layer is interpreted through AL and PIACIP attributes. The bottom interface of the weakly reinforced layer is determined through SIF and SRS attributes, and the subgrade defect areas are delineated by combining four types of GPR attributes. Source data are provided as a Source Data file.

with an effect equivalent to the SIF amplitude in Fig. 5c. The disordered areas of the attribute also clearly delineated the defective areas of the stratum (green irregular circles).

## Comparison between GPR results and borehole coring

Based on the above interpretation of GPR data, we selected two boreholes arranged near $C_2$ and $C_4$ in Fig. 4b for coring, which were marked as Borehole #1 (Fig. 6a, b) and Borehole #2 (Fig. 6c, d), respectively. And the depth of Borehole #1 and Borehole #2 were approximately 11.0 m and 10.0 m, respectively.

The coring results for Borehole #1 were divided into five segments (Fig. 6a, b). The first segment (0-2.37 m) and the third segment (5.92-8.25 m) corresponded to the 0-47.3 ns and 119.5-163.5 ns segments in the GPR attribute sections in Fig. 5, respectively. It was shown as gray cement block with good cementation and complete pile body. The second segment (2.37-5.92 m) corresponded to the 47.3-119.5 ns segment in the GPR attribute sections in Fig. 5. It was manifested as broken cement blocks with large pores and poor integrity of the pile body. This might be caused by the excessive acceleration of the nozzle during construction, resulting in uneven mixing of cement and soil. The fourth segment (8.25-9.69 m) corresponded to the 163.5-193.7 ns segment in the GPR attribute sections in Fig. 5. It appeared as gray cement blocks. The cementation property was not as good as that of the first segment. And the integrity of the pile body was slightly poor. The reason might be that the water content in the deep soil layer, resulting in poor pile formation effect of DCM piles. The fifth segment (9.69-11.0 m) was primary loess.

The coring results of Borehole #2 were divided into three segments (Fig. 6c, d). The first segment (0-4.91 m) corresponded to the 0-97.2 ns segment in the GPR attribute sections in Fig. 5, and the core sample characteristics were the same as those of the first and third segments of Borehole #1. The second segment (4.91-8.26 m) corresponded to the 97.2-164.5 ns segment in the GPR attribute sections in Fig. 5, and the core sample characteristics were the same as the fourth segment of Borehole #1. The third section (8.26-10.0 m) was primary loess.

It can be seen from the comparison between the above borehole coring results and the GPR attribute interpretation results that the two are basically consistent. Based on the drilling results and GPR interpretation results (Table 1), we further calculated the electromagnetic wave velocities. For the subgrade reinforced by DCM piles, there might be some differences in the physical properties of the medium due to the different construction quality at different locations, which might lead to some deviations in the electromagnetic wave velocities between them. However, this difference was relatively weak according to the statistical results of Borehole #1 and Borehole #2 (Table 1). Therefore, we took the average value of velocities ($v_{ave} \approx 0.1\,\text{m}\cdot\text{ns}^{-1}$) as the final electromagnetic wave velocity. And all GPR data processing results in Figs. 4, 5 were subjected to time-depth conversion processing based on this average velocity.

## Evaluation of ground improvement

Based on the interpretation results of typical channels, multi-channel section and attributes of GPR data (Fig. 6e) as well as constrained with boreholes (Fig. 6a, b, c, d), we had constructed a geological model (Fig. 6f) of the subgrade reinforced by DCM piles below this survey line. The stratum after ground improvement could be divided into three layers that were strongly reinforced layer, weakly reinforced layer and primary stratum. The average buried depth of the bottom interface of the strongly reinforced layer was approximately 6.0 m. The shallowest distributed on the right side of the section, approximately 4.4 m. And the deepest was distributed in the middle left of the section, approximately 8.5 m. The average buried depth of the bottom interface of the weakly reinforced layer was approximately 9.0 m. The shallowest was approximately 7.2 m, distributed in the middle right of the section. And the deepest was approximately 10.0 m, distributed in the middle left of the section. There were two defect areas in the stratum of the reinforced subgrade, which were identified as $D_1$ and $D_2$, respectively. The defect area $D_1$ presented an irregular shape and was distributed on the right side of the section. It had a large area, spanning of approximately 24.0 m and a buried depth of 2.1-6.6 m. The defect

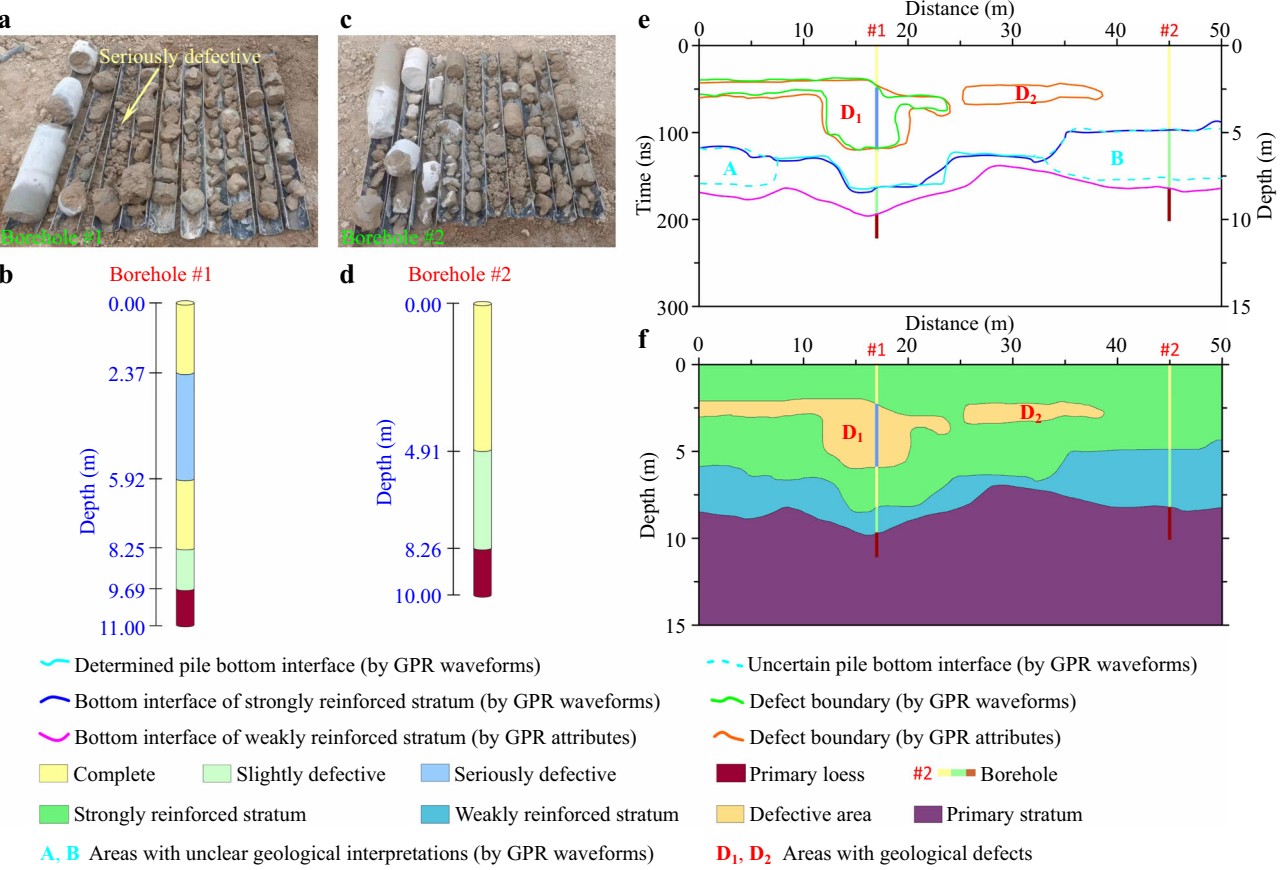

**Fig. 6 | Construction of geological model after ground improvement.**
**a, b, c, d** Drilling and coring results of Borehole #1 and Borehole #2. **e** Comparison of interpretation results of GPR waveforms (section) and attributes. **f** Geological model after ground improvement. The geological model is constructed based on the GPR interpretation results constrained by dilled results. The reinforced subgrade is divided into three layers that are strongly reinforced stratum, weakly reinforced stratum and primary stratum. There are two defects in the subgrade that are $D_1$ and $D_2$. Source data are provided as a Source Data file.

**Table 1 | Calculating electromagnetic wave propagation velocities by combining GPR interpretation and drilling results**

| Stratum interface | Borehole #1 | | | Borehole #2 | | |
|---|---|---|---|---|---|---|
| | Depth /m | Double-way travel time /ns | Velocity /m·ns⁻¹ | Depth /m | Double-way travel time /ns | Velocity /m·ns⁻¹ |
| Top interface of $D_1$ | 2.37 | 47.3 | 0.1002 | - | - | - |
| Bottom interface of $D_1$ | 5.92 | 119.5 | 0.0991 | - | - | - |
| Bottom interface of strongly reinforced stratum | 8.25 | 163.5 | 0.1009 | 4.91 | 97.2 | 0.1010 |
| Bottom interface of weakly reinforced stratum | 9.69 | 193.7 | 0.1001 | 8.26 | 164.5 | 0.1004 |
| Average velocity /m·ns⁻¹ | - | | 0.1 | - | | 0.1 |

area $D_2$ presented an approximately flat elliptical shape and distributed in the right side of the section. It spanned approximately 13.0 m, had a thickness of approximately 1.0 m and a buried depth of 2.2-3.3 m. Therefore, it was recommended to supplement piles to the defect $D_1$ and $D_2$ areas.

## Discussion

DCM piles have played a key role in the treatment of foundation settlement distress[17,19,21–24]. However, piling is a complex and systematic construction process. Small errors in each construction step may cause deviations between the pile body and the design, such as insufficient (or uneven) mixing of cement paste and deep soil particles as well as the lack of coordination between the lifting speed of the grouting pipe and the grouting amount, etc. Even if the pile construction process is carried out in accordance with standard process as well as the construction information such as the improvement length of each pile, pressure, speed, grout volume injected, etc., are recorded, defects in pile body and inconsistent pile lengths can not be avoided which creates potential safety hazards for highway engineering construction. This is why it is necessary to carry out pile foundation testing.

If there are defects in the pile foundations, the quality of ground improvement will inevitably decline. Therefore, there is a relationship of mutual influence and restriction between pile foundation defects and ground improvement quality. The ultimate purpose of pile foundation defect detection is to evaluate whether the quality of ground improvement has achieved good results. However, it is very challenging to accurately evaluate the quality of pile construction due to the

limitations of DCM pile material, large number of pile groups and small pile spacing. If the single pile detection technology is adopted, and then one pile after another is detected, not only will the detection effect be poor, but also the detection efficiency will be low, and the cost will be high. If the foundation strengthened by piles is regarded as a complete set of strata, the problem of pile foundation testing can be transformed into a problem of ground improvement quality evaluation. It can significantly reduce the detection cost while effectively improving the detection efficiency.

Piling is actually a process of transforming the original formation, and the transformed formation medium will undergo significant changes[14,59], which also will lead to changes in the electromagnetic properties of different underground locations. The degree of geological transformation varies, resulting in different changes in the underground medium and varying electromagnetic characteristics[55,56]. This creates conditions for using GPR technology to detect the changes in the stratum caused pile construction and the differences in the changes in the strata media between different underground locations. Difference in the electromagnetic properties of underground media can cause changes in the GPR response, specifically manifested in changes in the waveform, amplitude, frequency, phase and related attributes of the GPR signals[40,42,49,50,54,55]. However, the propagation and response of electromagnetic wave fields under the condition of pile groups are very complex, and it is not easy to fully reveal the propagation laws of electromagnetic waves under such condition. For this reason, we have constructed geological models of subgrade reinforcement by pile groups, and qualitatively discussed the electromagnetic wave propagation phenomenon in this case. Compared with the complex problem of pressure-controlled spherical cavity expansion in semi-infinite soil, although our models are simplified or idealized geological models, we believe that their GPR response mechanisms are identical. That is, GPR response is based on differences in the electromagnetic properties of media. As long as there are differences in underground media, differences in GPR response characteristics will inevitably be appeared. Therefore, GPR technology is fully applicable to the detection of geological defects caused by pressure-controlled spherical cavity expansion in semi-infinite soil. For the stationary oscillating waves appearing in the measured data (as shown in Fig. 4b), we speculate that this may be due to the guided wave phenomenon generated by electromagnetic waves within or between piles. We believe that these waves have a positive significance for judging the integrity of piles, and therefore can be considered as effective wave fields.

In theory, higher frequency and smaller spacing do contribute to improving the resolution of detection. However, if the frequency of GPR antennas is too high, the penetration depth of electromagnetic waves will be decreased. According to the existing technical experience of GPR[40-55], the maximum effective detection depth of 100 MHz antenna is approximately 50 m and the maximum effective detection depth of 250 MHz antenna is approximately 10 m in the survey of Quaternary strata. In addition, if the channel spacing is too small, it will inevitably increase the workload, thereby increase survey cost. In our case, we selected 100 MHz antennas for detection. The electromagnetic wave velocity $v$ obtained by matching GPR section with the drilled core was approximately $0.1 \, m \cdot ns^{-1}$ and the selected sampling rate $\triangle t$ was 0.351 ns. Therefore, the detection depth $h$ of 1024 sampling points per channel is approximately:

$$h = v \times \triangle t \times n/2 = 0.1 \times 0.351 \times 1024/2 = 17.97 m. \quad (1)$$

Our case shows that the data acquisition parameters, such as 100 MHz antenna, 2.0 cm point spacing, 0.351 ns sampling rate, and 1024 sampling points per channel, are fully suitable for the quality evaluation of soft soil subgrade reinforced by DCM piles with a length of approximately 10.0 m.

The signals that cause the GPR response will be very weak when the differences in the physical properties of underground media are quiet weak. This makes it difficult to identify the information of formation media differences from weak GPR signals[42]. It can be seen from Fig. 6e that the traditional waveform (section) interpretation results omit to identify the defect area $D_2$, and it is also difficult to accurately identify the bottom interface of the pile foundations. This is a manifestation of inaccurate interpretation results due to insufficient utilization of information carried by GPR data. Our research results show that the problem of fine quality evaluation of ground improvement can be effectively solved by integrating the GPR information of single-channel waveform, multi-channel section and attributes. On the basis of preliminary evaluation results obtained from waveform (section) interpretation, comprehensive analysis and interpretation of multi-attribute information can not only accurately identify the depth of DCM piles and subgrade defect areas, but also further subdivide the subgrade strengthened by pile groups into reinforced and weakly reinforced stratum. It provides an important technical guarantee for the safe and efficient construction of soft soil subgrade reinforcement projects.

The more data revealing a scientific or engineering problem is not necessarily the better, but whether it is typical and representative. Although we have only reported one GPR section, this GPR section has already included possible problems with DCM piles reinforcement of subgrade, such as deviation between pile length and design, defects in pile foundations, and differences in the reinforcement quality of subgrade in different sections. Therefore, our sample data fully possesses typical and representative characteristics. It should be emphasized that our work is mainly to detect the defects of DCM piles and evaluate the effectiveness of ground improvement based on GPR data. However, it is not rule out that there are other scientific issues in ground improvement engineering that can be studied or explored through GPR information. Multi-data samples may be an important way to discover and solve unknown scientific issues.

Each technology will have more or less deficiencies (or limitations), and so does our technology. Specifically, if there are strong electromagnetic field sources (such as high-voltage lines, substations, wireless communication base stations, etc.) or large metal objects (such as trucks, drilling machines, etc.) in the construction site environment, GPR technology will not achieve good detection results. Therefore, when using GPR survey, it is important to avoid these situations as much as possible. In addition, when the electromagnetic properties of the detection targets and the surrounding medium are close, it is easy to cause misjudgment, that is, GPR survey may not be applicable at all in this case. Moreover, GPR interpretation usually has multiple solutions. The GPR response characteristics caused by different media are close to each other or the same media causes different GPR response characteristics. What kind of interpretation can match the actual situation? The work experience of technicians will play an important role in this case. Finally, it should be noted that our GPR technology can effectively distinguish the electromagnetic differences of different formation media, but it is difficult to achieve a quantitative (or accurate) interpretation of the relationship between the ultimate expansion pressure and the ground surface displacement of the expansion pressure.

In summary, the effectiveness of soft soil subgrade reinforcement depends entirely on the quality of pile construction, and defect detection of pile foundations can be regarded as an effective monitoring measure for the quality of pile construction or as an insurance against accidents[33-39]. Therefore, in order to truly improve the quality of subgrade reinforcement by DCM piles, it is necessary to strictly follow the specifications during the pile construction process, eliminate all possible construction errors, and strictly control the construction quality. Cutting off the possibility of pile foundation defects from the source is the eternal goal pursued by subgrade engineering construction.

## Methods

GPR data processing and geological information interpretation are two key technologies that restrict the ability of GPR survey to achieve good geological results. We have established a workflow of GPR data processing and geological information interpretation for the defect detection of DCM piles and quality evaluation of ground improvement (Supplementary Fig. S3).

### GPR data processing

High quality GPR data processing is the foundation and guarantee for accurate and complete interpretation of geological information carried by GPR data[43]. If the quality of GPR data processing is not high, the geological information carried by GPR data will not be fully extracted or there may be omissions, and the effectiveness of pile foundation defect detection and ground improvement quality evaluation will be greatly reduced. According to the actual situation of defect detection of DCM piles and ground improvement quality evaluation as well as the characteristics of GPR data collected, in addition to zero correction, gain recovery, inter channel and intra channel energy compensation, it is necessary to emphasize processing such as interference noise suppression and multiple attenuation. The purpose is to ensure high-quality GPR data processing results without losing valid information. In addition, GPR data obtained under undulating surface conditions require topographic correction. Otherwise it will inevitably distort the GPR response characteristics caused by underground geological targets, leading to distortion of GPR interpretation results. Due to the relatively flat surface of our test site, no topographic correction was performed during GPR data processing in our case.

### Geological information interpretation

Only by fully mining the geological information carried in GPR data can we further accurately evaluate the quality of pile foundation and the effectiveness of ground improvement. The changes in GPR waveform, amplitude, frequency and phase are closely related to the physical mechanism of electromagnetic wave propagation, geotechnical physical properties, stratigraphic structure and other factors. GPR section interpretation is the interpretation of the geological information carried by GPR section based on the changes in waveform, amplitude, frequency and phase characteristics. It is the most basic content of GPR data interpretation[61]. Firstly, based on the theoretical GPR response characteristics and understanding obtained from the subgrade geological models of pile group reinforcement (Fig. 1), typical GPR channels in the section are extracted for response characteristics analysis (Fig. 4b), and information such as pile foundation complete and defects are initially obtained. Then, it is extended to GPR section interpretation (Fig. 4c) to obtain subgrade structure, defects and other information below the entire survey line based on the conclusion of single-channel interpretation.

The variation characteristics of waveform, amplitude, frequency and phase only reflect one aspect of geological information carried in GPR data. When underground reflection characteristics are obvious, they can effectively reflect the relationship between GPR response characteristics and stratum structure[61]. However, it is an indisputable fact that the structure and physical properties of the underground are complex and variable. When the difference in wave impedance is not significant, the characteristics changes in waveform, amplitude, frequency and phase of GPR data will be not significantly, but it does not indicate that the stratum structure or physical properties have not changed. Therefore, relying solely on the waveform, amplitude, frequency and phase characteristics of GPR data, it is often difficult to accurately reveal the true internal structural characteristics and physical property distribution of pile foundations and subgrade. On the other hand, in addition to the above four basic information, GPR data also contain rich attribute information that can be utilized. GPR attributes refer to extracting geometric, kinematic, dynamic and statistical characteristics related to electromagnetic wave response and propagation from GPR records. GPR attribute analysis is to extract descriptive and quantitative GPR attribute features to characterize the structure and physical property distribution information of underground targets[62,63]. Currently, GPR attribute information has been applied to distinguish rock stratum interface and fault fracture areas[64], detect light non aqueous phase liquids (LNAPL) pollution[65], identify collapsed ancient caves[66], high-resolution glacier imaging[67] and survey underground structure in archaeology[61,68,69]. These successful application cases show that extracting GPR attribute information for analysis can fully mine the geological information carried in GPR data, thereby enabling better interpretation of underground targets. Therefore, we have specially developed a GPR attribute information extraction and analysis technology for pile foundation defect detection and subgrade reinforcement quality evaluation. Here, we will briefly introduce the four most representative GPR attribute principles.

### Arc length of time window (AL)

The AL attribute is defined as the expanded length of the waveform curve within the analysis time window (Eq. 2). It is a joint attribute that combines amplitude and frequency characteristics and can be used to distinguish the phase characteristics of GPR records such as between strong amplitude and high frequency, between strong amplitude and low frequency, between weak amplitude and high frequency as well as between weak amplitude and low frequency (Fig. 5a).

$$AL = \frac{1}{n \cdot \triangle t} \sum_{i=1}^{n} \sqrt{[(A(i+1) - A(i)]^2 + \triangle t^2}, \qquad (2)$$

where, $n$ represents the number of sampling points within the analysis time window, $\Delta t$ forms the sampling rate (sampling time), $A(i)$ denotes the amplitude value of the $i^{th}$ sampling point, and $i$ is the sampling sequence number ($i = 1, 2, 3, ..., n$).

$\Delta t$ determines the resolution of the sampled signals. A smaller sampling time means a higher sampling resolution, but it also means an increase in sampling data. For GPR survey at subgrade scale, we recommend a sampling time of 0.1 to 0.4 ns. $\Delta t$ was taken 0.351 ns in our case. For $n$, statistical analysis is generally performed using sampling points with a half wavelet length. The smaller the length of the analysis time window, the higher the resolution of the GPR attribute section obtained. However, the length of the statistical analysis time window cannot be too small (such as $n < 2$), otherwise the statistical analysis will lose its significance. We took a statistical analysis time window with a length of 1.0 ns in our case, that is, took 2 data for each time window for statistical analysis.

**Product of instantaneous amplitude and cosine of instantaneous phase (PIACIP).** The PIACIP attribute strengthens the amplitudes of wave peaks and troughs, and inverts all trough amplitudes into apparent peak amplitudes (Fig. 5b), which is effective for analyzing amplitude anomalies.

$$PLACIP = A_{Inc}(t) \cdot \cos\left[Q_{Inc}(t)\right], \qquad (3)$$

where, $A_{Inc}$ represents instantaneous amplitude, $Q_{Inc}$ denotes instantaneous phase, and $t$ forms travel time, cos[] is the cosine function.

**Slope of instantaneous frequency (SIF).** The SIF attribute is defined as the change rate of instantaneous frequency over time within the analysis time window (Eq. 4). It highlights changes in local frequency, so it can more effectively reflect differences in thin layers, and is effective for dividing edge phases (Fig. 5c). It is also commonly used to

indicate the rate of attenuation and absorption.

$$SIF = \frac{df_{Inc}}{dt}, \tag{4}$$

where, $f_{Inc}$ denotes instantaneous frequency, $t$ forms travel time, and d represents derivation operator.

**Slope of reflection strength (SRS).** The SRS attribute is defined as the change of instantaneous reflection intensity (i.e. instantaneous amplitude) with reflection time (Eq. 5). If the reflection intensity remains throughout the analysis time window interval, the slope value will approach zero. If the reflection intensity gradually increases from the top to the bottom of the time window, the slope value is positive, on the contrary, the slope value is negative. The attribute can reflect the vertical distribution characteristics and interbedding conditions of the stratum (Fig. 5d).

$$SRS = \frac{A_{Ins}(t)}{dt}, \tag{5}$$

where, $A_{Ins}$ represents instantaneous amplitude, $t$ forms travel time, and d is derivation operator.

**Calculation of velocity**
It is necessary to know the propagation velocity of electromagnetic waves when conducting time-depth conversion for GPR section. Through joint calibration of drilling results and GPR section, the buried depth of the formation can be determined, and then the double-way travel time of the formation interface can be picked up from the GPR section to calculated the electromagnetic wave propagation velocity,

$$v = \frac{2h}{t}, \tag{6}$$

where, $v$ represents the propagation velocity of electromagnetic waves, $h$ forms the depth of the formation interface, and $t$ denotes the double-way travel time.

## Data availability
All the data supporting the findings of this study are available within the main text and the Supplementary Information. The source data generated in this study are provided in the Source Data file and also available in Figshare under accession code https://doi.org/10.6084/m9.figshare.22958399. Source data are provided with this paper.

## Code availability
The source codes and respective compiled executable are provided as Supplementary Software 1 file. The source code has also been made available in Figshare under accession code https://doi.org/10.6084/m9.figshare.22958399.

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

## Acknowledgements

The authors thank Professor Qingchun Li at Chang'an University, Professor Chengqian Tan and Professor Zhiping Zhang at Xi'an Shiyou University for valuable suggestions. The authors also thank the four anonymous reviewers for constructive comments and recommendations. H.S. thanks the Natural Science Basic Research Program of Shaanxi Province of China for Project 2017JZ007 and the Key Research and Development Project of Shaanxi Province of China for Project 2022GY-148.

## Author contributions

H.S., X.L., R.D. and Y.Z. devised the initial concept for the work. H.S., X.L. and R.D. designed the experiments. H.S., R.D. and Y.Y. developed the data processing software. R.D., X.L., Z.X., J.L., B.L., Ho.C. and J.G. collected the data. H.S., R.D., Y.Z., J.Z., Ha.C. and Y.Y. processed and analyzed the data. H.S., G.L. and Ha.C. drew the figures and table. H.S., X.L. and R.D. co-wrote the manuscript. H.S., X.L., R.D., Y.Z., J.Z., Ha.C., Z.X., C.Y., J.L., C.J. and Y.Y. discussed the results and commented on the manuscript at all stages.

## Competing interests

The authors declare no competing interests.
