## [Peer Review File · Nature Communications]

Quality evaluation of ground improvement by deep cement mixing piles via ground-penetrating radarREVIEWER COMMENTS

Reviewer #1 (Remarks to the Author):

This manuscript evaluates the quality of ground reinforced with high pressure jet grouting piles by the ground penetrating radar.

It should be remembered that there are many reports on techniques to evaluate the quality of ground reinforced with high pressure jet grouting piles in the world. However, this manuscript does not describe such technical information.

In addition, the reviewer feels that the manuscript is simply an advertisement for the ground penetrating radar developed by the authors.

This is because the principle of measurement by radar is not described in detail.

By the way, the ground penetrating radar developed by the authors penetrates the ground with a radar device into the ground, even though it is now required to evaluate the quality of ground reinforced with high pressure jet grouting piles by non-destructive testing. The reviewer was not aware of the technical limitations of the radar system. Therefore, the reviewers cannot perceive any technical novelty.

Reviewer #2 (Remarks to the Author):

The research addresses the quality evaluation of grouting pipes using ground penetrating radar with signal processing indices. The topic is interesting and impactful, and the results are valid. However, Additional discussions about the results and improvements of the writing may be needed (Q.9, Q.4, Q.10-14). Below are the comments:

Q.1 Abstract p.1 l.13-14, I am not clear about the differences between the defect detection of piles and subgrade reinforcement evaluation. Is the geological model the proposal?

Q.2 Introduction, important technical terms should be explained, for example p.3 l.47 loopholes, high strain/low strain dynamic measurement.

Q.3 Methodology p.7 l.131-134, the propagation phenomenon is discussed qualitatively and not verified at this point. Probably the phenomenon is too complicated to discuss. Stationary oscillating waves are found in the measurement data.

Q.4 p.11 l.208-210 many important gpr processing methods appear here and in my understanding some need delicate parameter selection but not shown. At least references are needed.

Q.5 I am a bit confused about the structure of the manuscripts. p.13 257-262 borehole coring and sec.3.4 signal attributes are the methodology.

Q.6 Engineering practice p.15 l.294-296, was the gpr device on the lime soil or just above the pile ?

Q.7 p.15 l.300-304 what is the target depth range of 1024 points and how did you select 100 MHz and spacing 2 cm? Do higher frequency or lower spacing help the interpretation and measurement? Are the parameters applicable to lossy soil?

Q.8 p.18 eq.1 n and delta t may be important but the information and how to select the parameters are missing.

Q.9 Fig.8 green and blue areas are surprisingly clear but how you pick up cyan lines is not clear. You may extract the end of alternating blue and red areas of piles. I guess eq.1-2 is related to energy and eq.3-4 is changing points. Then, how to select targets from many changing points is important.

Minor comments:

Q.10 p.5 l.88 "detect the quality" may be a strange expression.

Q.11 p.13 l.256 "kinds" may be types

Q.12 p.15 l.311 instead of circled 1-3 other symbols may avoid text garbling.

Q.13 p.16 l.315 "fig.2d" does not exist.

Q.14 p.19 l.384 "very" is subjective and not recommended.

Manuscript Review Summary

Manuscript Number: NCOMMS-23-05379

Title of paper: Quality evaluation of subgrade reinforced by high pressure jet grouting piles via ground penetrating radar

Article Type: Research Paper

Overview

The paper presents the application of GPR together with its interpretation in quality evaluation of the ground reinforced by grouting piles. The main idea is to consider the reinforced ground as a strong stratum which is different from traditional quality control that considers individual piles. The topic can be of interest to peer readers. However, there are several concerns about its technical errors and presentation, particularly the writing. With the present form of presentation, it is unsuitable for publication in this journal. The manuscript needs substantial revision to meet the standard of publishing in the journal. After correction, resubmission is encouraged. To improve the readability and contribution of the article, suggestions are provided as follows;

1. The main serious problem of this article is some technical errors as follows.
 - 1.1 The highlight of the work is the effectiveness of GPR in conjunction with the interpretation techniques. Instead of separately discussing each technique as detailed in Figure 8, the bottom interfaces and defective areas from each technique should be plotted together with the original ones (figure 7) and the results of drilling (figure 9). By doing this, more fruitful discussions on interpretation techniques can be done. In addition, the positions of piles should be also added in figures 7 and 9.
 - 1.2 In engineering practice, the construction information such as the improvement length of each pile, pressure, speed, grout volume injected, etc., are recorded. Why do the lengths of piles in this construction become unknown?
 - 1.3 Although the idea and methodology are good, only one section of the test is reported. The reliability of the work becomes questionable.
 - 1.4 No information on the quality control/assurance of the grouting pile construction conducted in the current practice is provided and discussed.

1.5 By using this idea, the reviewer agrees with the authors that the pile spacing must be small. Can the authors suggest the possible spacing? Perhaps from the theoretical point of view.

1.6 At the end of part 2.3.2, it seems that the GPR interpretation needs the coring results. This contradicts the state of the problem, the idea of using GPR for cost and time reduction.

2. Several modifications to manuscript preparation are necessary.

2.1 English is generally good, however, the writing is wordy. The manuscript needs to be more concise.

2.2 Please reconsider the meaning of several sentences, such as Lines 40-42, grouting piles are not low cost and green environmental protective. This is also the case for Lines 45-47.

2.3 Please reconsider several terminologies, such as, "safety accident", and "pile building".

2.4 Please reconsider the technical correctness of the writing, such as Lines 64-65. The concrete pile has not only cement and sand but also coarse aggregates.

2.5 The current writing style is rather a technical report. Please reconsider rewriting more comprehensively. In each part, please emphasize what has been done in the current work.

2.6 Conclusion needs rewriting, such as item 2.

2.7 Please reconsider the existence of Table 1.

Reviewer #4 (Remarks to the Author):

The authors submitted a paper on "Quality evaluation of subgrade reinforced by high-pressure jet grouting piles via ground penetrating radar", Overall, the manuscript is fascinating. It will add to the existing knowledge on ground improvement. The results of the study are of interest to the community. However, the manuscript needs revision before it can be accepted for publication. I will be glad to review the manuscript after the modifications are implemented.

1. Considering the scope and reputation of the journal, the manuscript in its current version can't be accepted for possible publication. In that regard, the authors must perform extensive revisions to the manuscript.

Abstract

2. The authors should clearly indicate the study's significance

3. in the abstract. In the current form, the importance is not shown.

- The authors should also highlight briefly the methodology employed in the study.
- A brief conclusion/significance of the study should also be indicated at the end of the abstract.
- In general, the authors are encouraged to improve the abstract further while keeping the journal's guidelines in mind.

Introduction

• The entire introduction should be revisited. The problem statement is not adequately highlighted. The novelty and gap the study is trying to fill are also unclear. The introduction needs a proper flow. The objective of the study should also be stated categorically. Add more previous studies in this area and their various shortcomings. The referencing format is also not correctly done. In line with the scope of the survey, you can use up to about 2.5 pages for the introduction.

- Line 65 sand + cement (+) is not appropriate.

Engineering Project

- In 3.1. Project Overview, here describe bore logs used for the study also the geology of the area; if possible, you can also provide some pictures of them
- A map for this subsection is also vital where the Jet grouting is located.

Results and Discussion

1. How is GPR suitable for the problem of pressure-controlled spherical cavity expansion in semi-infinite soil?

2. Would the measured ground heave be compared with the GPR predictions using the proposed theoretical model?

3. Can the author explain the limit expansion pressure and the expansion pressure-ground surface displacement relation?

4. Subsequently, the proposed approximate solutions are applied to interpret the limit injection pressure and the grouting pressure-ground surface displacement during the installation process of HPJG-GPR. Please clarify.

Dear reviewers,

Thank you very much for your constructive comments and recommendations. We have carefully revised the manuscript according to your suggestions and recommendations. Supplementary contents or major repairs have been marked with purple in the revised manuscript. I hope this edition can fulfill your demand. Our responses to comments and changes made in the manuscript are listed below. Please contact me if you have any other questions, suggestions or comments. Thanks again!

Sincerely regards,

The authors

Reviewer: 1

(1) It should be remembered that there are many reports on techniques to evaluate the quality of ground reinforced with high pressure jet grouting piles in the world. However, this manuscript does not describe such technical information.

Response: Thank you for your good advice! We have added some references and reports on techniques to evaluate the quality of ground reinforced with high pressure jet grouting (HPJG) piles in the revised manuscript according to your suggestion, and described such technical information.

(2) The reviewer feels that the manuscript is simply an advertisement for the ground penetrating radar developed by the authors. This is because the principle of measurement by radar is not described in detail.

Response: Thank you for your suggestion! As GPR is a mature and widely used technology, there are a lot of documents about this technology. Therefore, we only briefly introduce the principle of measurement by GPR in the manuscript to avoid creating the illusion of lengthy discussion for readers. It should be noted that the focus of our work is how to use the information carried by GPR data to effectively evaluate the quality of subgrade reinforcement by HGJP piles.

(3) The ground penetrating radar developed by the authors penetrates the ground with a radar device into the ground, even though it is now required to evaluate the quality of ground reinforced with high pressure jet grouting piles by non-destructive testing. The reviewer was not aware of the technical limitations of the radar system.

Response: Thank you very much for your suggestion! We agree with you. Each technology will have more or less deficiencies (or limitations), and so does our technology. Specifically, if there are strong electromagnetic field sources (such as high-voltage lines, substations, wireless communication base stations, etc.) or large metal objects (such as trucks, drilling machines, etc.) in the construction site environment, GPR technology will not achieve good detection results. Therefore, it is important to avoid these situations as much as possible when using GPR survey. In addition, it is easy to cause misjudgment when the electromagnetic properties of the detection targets and the surrounding medium are close, that is, GPR survey may not be applicable at all in this case. Moreover, GPR interpretation usually has multiple solutions. The GPR response characteristics caused by different media are close to each other or the same media causes different GPR response characteristics. What kind of interpretation can match the actual situation? The work experience of technicians will play an important role in this case.

Reviewer: 2

(1) Q.1 Abstract p.1 l.13-14, I am not clear about the differences between the defect detection of piles and subgrade reinforcement evaluation. Is the geological model the proposal?

Response: Thank you for your question! If the pile foundation has defects, the quality of subgrade reinforcement will inevitably decline. Therefore, there is a relationship of mutual influence and restriction between pile foundation defects and subgrade reinforcement quality. In fact, the ultimate purpose of pile foundation defect detection is to judge whether the quality of subgrade reinforcement has achieved good results.

However, it is very challenging to accurately evaluate the quality of pile construction due to the limitations of pile material, large number of pile groups and small pile spacing. If the single pile detection technology is adopted, and then one pile after another is detected, not only will the detection effect be poor, but also the detection efficiency will be low, and the cost will be high. If the subgrade strengthened by piles is regarded as a complete set of strata, the problem of pile foundation detection can be transformed into the problem of subgrade reinforcement quality evaluation. The purpose of this is to create conditions for the selection of continuous detection technology, and try to significantly reduce the detection cost while effectively improving the efficiency and effect of pile foundation detection. Therefore, the geological models in Figure 1 constructed by us are indeed a proposal to solve the above problems.

(2) Q.2 Introduction, important technical terms should be explained, for example p.3 l.47 loopholes, high strain/low strain dynamic measurement.

Response: Thank you very much for your suggestion! We have supplemented the interpretation of these important technical terms in the revised manuscript according to your suggestion.

(3) Q.3 Methodology p.7 l.131-134, the propagation phenomenon is discussed qualitatively and not verified at this point. Probably the phenomenon is too complicated to discuss. Stationary oscillating waves are found in the measurement data.

Response: We fully agree with you. The propagation and response of electromagnetic wave field under pile group conditions are indeed very complex, and it is very challenging to fully reveal the propagation law of electromagnetic wave under such conditions. In the manuscript, we qualitatively discussed the electromagnetic wave propagation phenomenon in this case based on our knowledge (such as geometry). For the stationary oscillating waves in the measured data, we speculate that it may be the guided wave phenomenon generated by the electromagnetic wave in pile or between piles. We believe that this guided wave has positive significance in judging

the integrity of piles, so it can be regarded as an effective wave fields.

(4) Q.4 p.11 l.208-210 many important gpr processing methods appear here and in my understanding some need delicate parameter selection but not shown. At least references are needed.

Response: Thank you for your good suggestion! It is an indisputable fact that it is necessary to select reasonable processing parameters to ensure the quality of data processing in the process of GPR data processing. In the revised manuscript, we have appropriately supplemented relevant contents according to your suggestions.

(5) Q.5 I am a bit confused about the structure of the manuscripts. p.13 257-262 borehole coring and sec.3.4 signal attributes are the methodology.

Response: Thank you for your good advice! With regard to the content of p.13 257-262 in the original manuscript, our purpose is to further explain the data processing and interpretation process, while the content of sec. 3.4 is a detailed explanation of the project case. In order to avoid confusion, we have made appropriate revisions to the relevant contents in the revised manuscript.

(6) Q.6 Engineering practice p.15 l.294-296, was the gpr device on the lime soil or just above the pile?

Response: Yes, GPR device was on the lime soil layer.

(7) Q.7 p.15 l.300-304 what is the target depth range of 1024 points and how did you select 100 MHz and spacing 2 cm? Do higher frequency or lower spacing help the interpretation and measurement? Are the parameters applicable to lossy soil?

Response: Thank you for your question! According to the existing technical experience of GPR survey, the maximum detection depth of 100 MHz antenna is approximately 50 m and the maximum detection depth of 250 MHz antenna is less than 10 m for conventional Quaternary sedimentary strata. In our case, the design parameters of pile foundation generally include pile diameter of 0.5 m, pile spacing of 1.0 m and pile length is 8.0 m. And the electromagnetic wave velocity v was

approximately 0.1 m/ns obtained by matching the drill core with GPR section and the selected sampling rate Δt was 0.351 ns. Therefore, the detection depth of 1024 points was approximately $h=v\times\Delta t\times n/2=0.1\times 0.351\times 1024/2= 17.97$ m. In order to ensure the lateral accuracy of detection, the point spacing should be selected as small as possible. Therefore, it is completely suitable in the quality evaluation of soft soil subgrade reinforcement to select 100~200 MHz antennas and approximately 2 cm point spacing. In theory, higher frequency and smaller spacing do help to improve the resolution or accuracy of detection. However, if the frequency of GPR antenna is too high, it will inevitably lead to the reduction of the depth of electromagnetic wave energy penetrating stratum. When the frequency of GPR antenna is high to a certain extent, it will not be able to detect the bottom interface of the pile foundation. In addition, if the point spacing is too small, it will inevitably increase the workload and thus increase the survey cost. Our case showed that the GPR data acquisition parameters we selected were fully applicable to the detection of damaging soil.

(8) Q.8 p.18 eq.1 n and delta t may be important but the information and how to select the parameters are missing.

Response: Thank you very much for your suggestion! We agree with you. These parameters determine whether the extracted GPR attribute information is true and reliable. In GPR attribute analysis, Δt represents sampling rate or sampling time, and n represents the number of sampling points in a analysis window. Δt determines the resolution of the sampled signal. The smaller the sampling time is, the higher the sampling resolution is. But it also means the increase of data. For GPR survey at subgrade scale, we recommend the sampling time of 0.1~0.4 ns. In our case, the sampling time was 0.351 ns. For n , generally take the number of sampling points with half wavelet length for statistical analysis. Of course, the smaller the analysis time window length, the higher the resolution of the GPR attribute section obtained. But the time window length of statistical analysis cannot be too small (such as $n<2$), otherwise the statistical analysis will lose its significance. In our case, we took the length of statistical analysis time window as 1 ns, that is, take 2 points for each time

window for statistical analysis. We have added relevant information in the revised manuscript.

(9) Q.9 Fig.8 green and blue areas are surprisingly clear but how you pick up cyan lines is not clear. You may extract the end of alternating blue and red areas of piles. I guess eq.1-2 is related to energy and eq.3-4 is changing points. Then, how to select targets from many changing points is important.

Response: We agree with your understanding! Different media have different responses to electromagnetic waves. Our research shows that GPR attribute information analysis can effectively identify the interface of media differences, such as energy differences and changes in response signals. However, when the difference of medium physical properties is quite weaker, the resulting GPR response signal is also weaker, which makes it difficult to identify the interface of formation medium difference in the weak GPR signal. For this problem, our strategy is to integrate multiple GPR attribute information to determine. For example, in the revised manuscript, the interface identified by the cyan line in Figure 5 is obtained by integrating the two GPR attribute information in Figure 5c and d. The basic principle of recognition is to take the dividing point between high frequency and low frequency of attribute signals.

(10) Q.10 p.5 l.88 "detect the quality" may be a strange expression.

Response: This is a good suggestion! We have dealt with it according to your suggestion, and correct it.

(11) Q.11 p.13 l.256 "kinds" may be types.

Response: Thank you very much for your suggestion! We have processed it in the revised manuscript according to your suggestion, and correct it to "types".

(12) Q.12 p.15 l.311 instead of circled 1-3 other symbols may avoid text garbling.

Response: Thank you for your good suggestions. We have processed them in the revised manuscript according to your suggestions.

(13) Q.13 p.16 l.315 "fig.2d" does not exist.

Response: Thank you for your suggestion! This is a mistake caused by our negligence. We have corrected it to "Fig. 1e" in the revised manuscript.

(14) Q.14 p.19 l.384 "very" is subjective and not recommended.

Response: Thank you very much for your suggestion! We have deleted it in the revised manuscript.

Reviewer: 3

(1) The highlight of the work is the effectiveness of GPR in conjunction with the interpretation techniques. Instead of separately discussing each technique as detailed in Figure 8, the bottom interfaces and defective areas from each technique should be plotted together with the original ones (figure 7) and the results of drilling (figure 9). By doing this, more fruitful discussions on interpretation techniques can be done. In addition, the positions of piles should be also added in figures 7 and 9.

Response: This is a very good suggestion! We have added relevant contents in the revised manuscript and made further discussion. However, the density of pile groups is large, and it is difficult to show a clear visual effect if the positions of piles are embedded in Figures 7 and 9 of the original manuscript. Therefore, we only added the drilling positions in these two figures.

(2) In engineering practice, the construction information such as the improvement length of each pile, pressure, speed, grout volume injected, etc., are recorded. Why do the lengths of piles in this construction become unknown?

Response: Thank you very much for your question! Pile building is a complex and systematic construction process. A small mistakes in each construction link may cause the deviation between the pile body and the design, such as the insufficient mixing (or uneven mixing) of the jet slurry and the soil particles cut by jet, and the lack of coordination between the lifting speed and rotation speed of the grouting pipe and the amount of jet grouting. Even if the pile construction process is carried out according

to the standard process, the defects of the pile body and the inconsistent pile length can not be avoided. Because of this, it is necessary to carry out pile foundation inspection. Our research can also be regarded as an important pile construction quality monitoring measure.

(3) Although the idea and methodology are good, only one section of the test is reported. The reliability of the work becomes questionable.

Response: Thank you for your question! And we can also understand your concern. In fact, the more data revealing a scientific or engineering problem is not necessarily the better, but whether it is typical and representative. Although we have only reported one GPR section, this GPR section has included the problems that may occur when HPJG piles are used to reinforce the subgrade, such as the deviation between the pile length and the design, defects in the pile foundation, and differences in the quality of subgrade reinforcement in different sections. Therefore, our sample data are completely typical and representative, and the results obtained from the research are completely trustworthy. It should be emphasized that our work is mainly to detect the defects of HPJG piles and evaluate the effect of subgrade reinforcement based on GPR information, but it does not rule out that there are other scientific problems that can be studied or explored through GPR information in the subgrade reinforcement project. Multi-data samples may be an important way and means to discover and solve unknown scientific problems.

(4) No information on the quality control/assurance of the grouting pile construction conducted in the current practice is provided and discussed.

Response: This is a very good suggestion! We have added relevant content in the discussion of the revised manuscript. In fact, the effect of subgrade reinforcement depends entirely on the quality of pile construction, and pile foundation defect detection can be regarded as a monitoring measure of pile construction quality. Therefore, in order to really improve the quality of subgrade reinforcement by HPJG piles, during the pile construction process, it is necessary to strictly follow the specifications to prevent all possible construction mistakes, and strictly control the

construction quality. Cutting off the possibility of pile foundation defects from the source is the eternal goal of subgrade reinforcement engineering.

Reviewer: 4

(1) Considering the scope and reputation of the journal, the manuscript in its current version can't be accepted for possible publication. In that regard, the authors must perform extensive revisions to the manuscript.

Response: Thank you very much for your suggestions and comments! We have revised it according to your suggestions.

(2) In the Abstract. The authors should clearly indicate the study's significance.

Response: This is a very good suggestion. We have carefully rewritten the abstract and added relevant contents in the revised manuscript according to your suggestion.

(3) In the abstract. In the current form, the importance is not shown. The authors should also highlight briefly the methodology employed in the study. A brief conclusion/significance of the study should also be indicated at the end of the abstract. In general, the authors are encouraged to improve the abstract further while keeping the journal's guidelines in mind.

Response: This is a very good suggestion! We have further improved the abstract in the revised manuscript according to your suggestion.

(4) Introduction. The entire introduction should be revisited. The problem statement is not adequately highlighted. The novelty and gap the study is trying to fill are also unclear. The introduction needs a proper flow. The objective of the study should also be stated categorically. Add more previous studies in this area and their various shortcomings. The referencing format is also not correctly done. In line with the scope of the survey, you can use up to about 2.5 pages for the introduction. Line 65 sand + cement (+) is not appropriate.

Response: Thank you very much for your comments and suggestions! We have improved these contents in the revised manuscript according to your suggestions.

(5) Engineering Project. In 3.1. Project Overview, here describe bore logs used for the study also the geology of the area; if possible, you can also provide some pictures of them. A map for this subsection is also vital where the Jet grouting is located.

Response: Thank you very much for your suggestion! We have added the location map of the work area and borehole coring photos in the revised manuscript.

(6) How is GPR suitable for the problem of pressure-controlled spherical cavity expansion in semi-infinite soil?

Response: This is a good question! In the manuscript! We built geological models of pile groups to strengthen the subgrade and discuss their GPR response characteristics. Compared with the complex problem of pressure-controlled spherical cavity expansion in semi-infinite soil, although the models we built are simplified or idealized geological models, we believe that their GPR response mechanism is completely the same, that is, GPR response is based on the difference of electromagnetic properties of media. As long as there are differences in underground media, the difference in GPR response characteristics will inevitably occur. Therefore, GPR is fully applicable to the detection of geological defects caused by pressure-controlled spherical cavity expansion in semi-infinite soil.

(7) Would the measured ground heave be compared with the GPR predictions using the proposed theoretical model?

Response: Thank you for your question! The GPR data obtained from the GPR survey under the condition of ground undulation need to be topographic corrected, otherwise it will inevitably distort the GPR response characteristics caused by the underground geological targets, thus leading to the distortion of the GPR interpretation results. Since the surface of our test site is relatively flat, no topographic correction is performed during GPR data processing. In addition, the theoretical models we put forward is not to match the actual geological conditions, but to reveal the GPR response characteristics in the case of geological defects and guide the interpretation of geological information carried by GPR data. Therefore, there is no

comparability between the two.

(8) Can the author explain the limit expansion pressure and the expansion pressure-ground surface displacement relation?

Response: Thank you for your question! The process of pile construction will cause expansion and extrusion to the stratum, and the resulting extrusion pressure will cause ground surface displacement. In fact, this process has also led to changes in the electromagnetic properties of different underground locations. The ground surface displacement caused by different expansion pressures is different, and the resulting electromagnetic properties are also different. This creates conditions for using GPR technology to detect what changes have taken place in the stratum due to pile construction and the differences in the displacement changes between different locations on the ground surface. However, our GPR technology can effectively distinguish the electromagnetic differences of different formation media, but it is difficult to explain the quantitative (or accurate) relationship between the limit expansion pressure and the expansion pressure ground surface displacement.

(9) Subsequently, the proposed approximate solutions are applied to interpret the limit injection pressure and the grouting pressure-ground surface displacement during the installation process of HPJG-GPR.

Response: Thank you for your suggestion! Through question 8, we have answered that GPR technology can effectively distinguish the electromagnetic differences of different formation media. The electromagnetic difference of the medium will cause the GPR response to change, which is embodied in the waveform, amplitude, frequency, phase and related GPR attributes of GPR data (or signals). Therefore, on the GPR section, the boundary of grouting pressure-ground surface displacement caused by different limit injection pressure can be explained by identifying and tracking the GPR response characteristic difference boundary points on different channels.

REVIEWER COMMENTS

Reviewer #2 (Remarks to the Author):

I appreciate your responses and modifications. The manuscript is much improved compared with the previous one. It seems the discussions and visibility of the result figures are also improved. I expect the manuscript for future publication.

Reviewer #3 (Remarks to the Author):

Manuscript Review Summary: review-NCOMMS-23-05379_R1:

Title of paper: Quality evaluation of subgrade reinforced by high pressure jet grouting piles via ground penetrating radar

Comments:

The comments on article preparation and presentation have been either answered and/or changed per my suggestions. There are, however, comments on the technical depth that are not completely addressed; these are extracted from the original review comments.

Main Issue

In the original comments, items 1.1-1.4 on the first page and 1.5-1.6 and 2.1-2.7 on the second page. In the revision, only items 1.1-1.4 have been addressed. Please recheck the file of the original review from reviewer #3 and address all comments.

Original comments

1. In engineering practice, the construction information such as the improvement length of each pile, pressure, speed, grout volume injected, etc., are recorded. Why do the lengths of piles in this construction become unknown?

Authors' Response: Thank you very much for your question! Pile building is a complex and systematic construction process. A small mistakes in each construction link may cause the deviation between the pile body and the design, such as the insufficient mixing (or uneven mixing) of the jet slurry and the soil particles cut by jet, and the lack of coordination between the lifting speed and rotation speed of the grouting pipe and the amount of jet grouting. Even if the pile construction process is carried out according to the standard process, the defects of the pile body and the inconsistent pile length can not be avoided. Because of this, it is necessary to carry out pile foundation inspection. Our research can also be regarded as an important pile construction quality monitoring measure.

Comments: The original comment is not for the necessity of quality assurance. In the installation of jet grouting piles, the maximum penetrated length of the jetting tube can be recorded on-site for every pile. The pile length of every installed pile is thus known.

2. Another main problem of this paper is **its presentation**, several result presentation and preparation do not reach publication standards. Although improvements based on the suggestions of other reviewers have been done, there are still several corrections. To minimize

the review time, please pay more attention to the revision. The following suggestions are given for revision;

- Please **use correct terminology**, such as
 - o Line 271; “When the ground was consolidated...”.
- Line 275; “but the behavior matched the consolidation and bending of the structure using kdist.”
- With the small difference between dry and saturated density values in Table 2, it is doubted.
- Several result explanations need to be reconsidered, such as

Authors’ Response: some corrections have been made and

Please note that the material is river gravel, which is almost impermeable material (water absorption lower than 1.0 %). Please See: M. A Da Silva et al. (2017), Rheological and mechanical behavior of High Strength Steel Fiber-River Gravel Self Compacting Concrete, Construction and Building Materials 150:606-618, 10.1016/j.conbuildmat.2017.06.030.

Comments: Please reconsider using the use of terms “consolidated” and “consolidation”. Please also reconsider that Table 2 lists properties of “aggregate” or “concrete slab”? Please make it clear.

Reviewer #4 (Remarks to the Author):

The authors have made a significant effort to improve the manuscript quality. In the current technical state, I have no more comments. However, I will recommend that the manuscript goes through extensive proofreading to correct grammatical, spelling, and contextual errors.

Dear reviewers,

Thank you very much for your constructive comments and recommendations. We have carefully revised the manuscript according to your suggestions and recommendations.

In addition, in order to accurately answer the questions you raised and convey the central idea and innovation of our paper to the readers, we once again contacted the pile construction engineers during the manuscript revision process to understand the entire process of pile construction. After in-depth communication with them, we discovered a problem that we mistook the deep cement mixing (DCM) pile for the high-pressure jet grouting (HPJG) pile. With respect for scientific facts and a responsible attitude, we have corrected the HPJG pile to DCM pile in this manuscript revision, and also updated relevant references. Although HPJG piles and DCM piles belong to two different types, we believe that their principles of defect detection and quality evaluation of roadbed reinforcement are similar. More importantly, the materials of these two types of piles are similar. Therefore, our technology is applicable to the detection of HPJG piles and DCM piles, as well as the quality evaluation of ground improvement.

Supplementary contents or major repairs have been marked in the revised manuscript. I hope this edition can fulfill your demand. Our responses to comments and changes made in the manuscript are listed below. Please contact me if you have any other questions, suggestions or comments. Thanks again!

Sincerely regards,

The authors

Reviewer #2

(1) I appreciate your responses and modifications. The manuscript is much improved compared with the previous one. It seems the discussions and visibility of the result figures are also improved. I expect the manuscript for future publication.

Response: We greatly appreciate your positive assessment and constructive suggestions on our manuscript.

Reviewer #3

(1) In the original comments, items 1.1-1.4 on the first page and 1.5-1.6 and 2.1-2.7 on the second page. In the revision, only items 1.1-1.4 have been addressed. Please recheck the file of the original review from reviewer #3 and address all comments.

Response: We greatly appreciate your constructive suggestions on our manuscript, and we sincerely apologize for our negligence! We have carefully addressed the remaining comments, as follows.

1.5 By using this idea, the reviewer agrees with the authors that the pile spacing must be small. Can the authors suggest the possible spacing? Perhaps from the theoretical point of view.

Response: Thank you for your question. In theory, there may be differences in the bearing capacity of different soft soil subgrade. From the perspective of reinforcement effect and cost saving in pile construction, there will inevitably be differences in the pile spacing used. Due to our professional background not being in pile engineering, we are unable to provide accurate pile spacing parameters. However, we believe that a feasible approach is to solve this problem through numerical simulation.

1.6 At the end of part 2.3.2, it seems that the GPR interpretation needs the coring results. This contradicts the state of the problem, the idea of using GPR for cost and time reduction.

Response: This is a good question. We emphasize that 1~2 boreholes can be properly arranged on the GPR survey line (or network), rather than many boreholes. The purpose of borehole coring is not only to verify the GPR interpretation results, but also to calibrate the GPR section with borehole results to further calculate and obtain more accurate electromagnetic wave velocities to achieve time-depth conversion

processing of the GPR section.

2.1 English is generally good, however, the writing is wordy. The manuscript needs to be more concise.

Response: Thank you very much for your good suggestion. We have revised the manuscript according to your suggestion.

2.2 Please reconsider the meaning of several sentences, such as Lines 40-42, grouting piles are not low cost and green environmental protective. This is also the case for Lines 45-47.

Response: Thank you for your good suggestion. We have revised them according to your suggestion.

2.3 Please reconsider several terminologies, such as, “safety accident”, and “pile building”.

Response: Thank you for your good suggestion. We have revised them according to your suggestion.

2.4 Please reconsider the technical correctness of the writing, such as Lines 64-65. The concrete pile has not only cement and sand but also coarse aggregates.

Response: Thank you for your good suggestion. We have revised it according to your suggestion.

2.5 The current writing style is rather a technical report. Please reconsider rewriting more comprehensively. In each part, please emphasize what has been done in the current work.

Response: This is a good suggestion. We have revised them according to your suggestion.

2.6 Conclusion needs rewriting, such as item 2.

Response: Thank you for your good suggestion. We have revised it according to your suggestion.

2.7 Please reconsider the existence of Table 1.

Response: Thank you for your suggestion. We have deleted Table 1 from the original manuscript in the revised manuscript.

(3) In engineering practice, the construction information such as the improvement length of each pile, pressure, speed, grout volume injected, etc., are recorded. Why do the lengths of piles in this construction become unknown?

Response: This is a very good question. It is indeed possible to use monitoring instruments to record the length of the grouting pipe entering the formation in the construction process, as well as other construction information such as grouting pressure and grouting volume. However, there may be differences in physical properties, structure, pressure, water content, and other factors between different strata. In addition, during the pile construction process, there may be phenomena such as insufficient mixing of cement slurry and deep soil particles, and a lack of reasonable coordination between the lifting speed of the grouting pipe and the grouting amount. The above factors can lead to incomplete pile formation in the formation after grouting that result in inconsistency between the actual pile length and the designed pile length.

(4) Another main problem of this paper is its presentation, several result presentation and preparation do not reach publication standards. Although improvements based on the suggestions of other reviewers have been done, there are still several corrections. To minimize the review time, please pay more attention to the revision. The following suggestions are given for revision;

Please use correct terminology, such as Line 271; “When the ground was consolidated...”.

- Line 275; “but the behavior matched the consolidation and bending of the structure using kdist.”

- With the small difference between dry and saturated density values in Table 2, it is doubted.

- Several result explanations need to be reconsidered, such as

Response: Thank you very much for your suggestions and opinions. However, when

we carefully examined our original manuscript, we found that it seemed that the detailed modification suggestions you provided were not applicable to our manuscript. Nevertheless, we have repeatedly proofread the manuscript, which mainly includes adjusting some sentences and correcting some grammar errors.

Reviewer #4

(1) The authors have made a significant effort to improve the manuscript quality. In the current technical state, I have no more comments. However, I will recommend that the manuscript goes through extensive proofreading to correct grammatical, spelling, and contextual errors.

Response: We greatly appreciate your positive assessment of our manuscript. And thank you also for your good suggestions. We have repeatedly proofread the manuscript.

REVIEWERS' COMMENTS

Reviewer #3 (Remarks to the Author):

The authors have addressed the reviewer's main points and that the paper is now ready to be published.

Dear reviewers,

Thank you very much for your constructive comments and recommendations.

Sincerely regards,

The authors